# Protective Effect of Astaxanthin on Ochratoxin A-Induced Kidney Injury to Mice by Regulating Oxidative Stress-Related NRF2/KEAP1 Pathway

**DOI:** 10.3390/molecules25061386

**Published:** 2020-03-18

**Authors:** Lin Li, Yueli Chen, Danyang Jiao, Shuhua Yang, Lin Li, Peng Li

**Affiliations:** Key Laboratory of Zoonosis of Liaoning Province, College of Animal Science & Veterinary Medicine, Shenyang Agricultural University, Shenyang 110866, China; 2017200134@stu.syau.edu.cn (L.L.); 2018220508@stu.syau.edu.cn (Y.C.); 2018240365@stu.syau.edu.cn (D.J.)

**Keywords:** astaxanthin, ochratoxin A, oxidative damage, NRF2, KEAP1

## Abstract

The present study aimed to investigate the effects of astaxanthin (ASX) on ochratoxin A (OTA)-induced renal oxidative stress and its mechanism of action. Serum kidney markers, histomorphology, ultrastructural observation, and oxidative stress indicators were assessed. Meanwhile, quantitative real-time reverse transcription PCR and western blotting detection of NRF2 (encoding nuclear factor, erythroid 2 like) and members of the NRF2/KEAP1 signaling pathway (KEAP1 (encoding Kelch-like ECH-associated protein), NQO1 (encoding NAD(P)H quinone dehydrogenase), HO-1 (encoding heme oxygenase 1), γ-GCS (gamma-glutamylcysteine synthetase), and GSH-Px (glutathione peroxidase 1)) were performed. Compared with the control group, the OTA-treated group showed significantly increased levels of serum UA (uric acid) and BUN (blood urea nitrogen), tubular epithelial cells were swollen and degenerated, and the levels of antioxidant enzymes decreased significantly, and the expression of NRF2 (cytoplasm), NQO1, HO-1, γ-GCS, and GSH-Px decreased significantly. More importantly, after ASX pretreatment, compared with the OTA group, serum markers were decreased, epithelial cells appeared normal; the expression of antioxidant enzymes increased significantly, NQO1, HO-1, γ-GCS and GSH-Px levels increased significantly, and ASX promoted the transfer of NRF2 from the cytoplasm to the nucleus. These results highlight the protective ability of ASX in renal injury caused by OTA exposure, and provide theoretical support for ASX’s role in other mycotoxin-induced damage.

## 1. Introduction

Ochratoxin is an important secondary metabolite and mycotoxin produced by *Aspergillus* and *Penicillium* species, which was first purified from *Aspergillus* culture [1]. There are three main forms of ochratoxin in nature: ochratoxin A (OTA), B (OTB), and C (OTC). The International Agency for Research on Cancer classified OTA as 2B (potential human carcinogen) [2]. Among the ochratoxins, OTA is the most widespread and most potent type, and pollutes agricultural products and affects human health [2,3]. OTA was first discovered in corn and later detected in grains and soybeans [4]. A variety of plant products and foods, such as cereals, fruits, wine, beer, coffee, cocoa and chocolate, Chinese herbal medicines and seasonings can be contaminated with OTA [5,6]. OTA pollution of animal feed is also a serious problem. When animals eat OTA-contaminated feed, OTA accumulates in the body, and is not easily detoxified by the animal’s metabolism. Therefore, OTA is often detected in animal products, especially in pig (kidneys, liver, muscles, and blood) and dairy products [7]. A variety of toxic effects are induced by OTA, such as carcinogenicity, teratogenicity, potential endocrine disruption, and immunotoxicity, particularly in humans and animals, and is thought to be closely related to human Balkan endemic nephropathy [8,9,10,11]. OTA triggers reactive oxygen species (ROS) production and leads to oxidative stress through a variety of direct and indirect mechanisms [12]. Furthermore, genetic toxicology studies provided strong evidence, for OTA-mediated mechanisms, of oxidative damage [13,14]. OTA significantly reduces the expression of genes involved in oxidative stress in the kidney tissues, and many of these genes contain antioxidant regulatory elements (AREs) in their promoter regions [15]. Antioxidant regulatory elements are identified by the transcription factor NF-E2-related factor 2 (NRF2), which regulates genes encoding detoxification, cell protection, and antioxidant enzymes [16]. In addition, OTA weakens the cell’s antioxidant defense barrier, making cells more susceptible to oxidative damage [17].

Natural astaxanthin (ASX) belongs to the lutein family, and is also known as lobster shell pigment. ASX is a beta-carotene found in shrimp and crab shells, salmon, oysters, algae, and fungi [18]. Studies have shown that ASX has biological activities that include anti-oxidation, anti-tumor, prevention of cardiovascular diseases, improvement of immunity, and coloration [19]. The molecular structure of ASX determines its ability to effectively quench singlet oxygen and scavenge free radicals [20]. ASX has a strong antioxidant activity and is known as a “super antioxidant” [21]. ASX is the only carotenoid that can pass through the blood–brain barrier [22]. Its antioxidant ability is the strongest of all carotenoids [23]. Furthermore, ASX can effectively prevent oxidative damage to human tissues, organs, cells, and DNA, thereby ameliorating the signs of aging caused by oxidation [24]. ASX can enhance the antioxidant capacity of elderly humans, thus delaying their natural aging [25]. ASX could maintain mitochondrial function, thus preventing bisphenol A-induced renal toxicity in rats [26]. In addition, ASX protects the skin from UVA light-induced oxidation [27]. These biological activities endow ASX with high economic value as a premium health care product, pharmaceutical, and feed additive [28,29].

Many studies have shown that after OTA enters the body, it causes damage to multiple tissues such as the kidney [30], liver [31], spleen [32], intestine [33], lung [34], and brain [35]. Therefore, the present study aimed to investigate whether ASX could protect kidney tissue from OTA-induced oxidative stress through the NRF2 pathway and to determine the protective effect of ASX on OTA toxicity in C57BL/J mice treated with OTA.

## 2. Results

### 2.1. Average Body Weight, Organ Coefficients, and Changes in Serum Biochemical Indicators

In the OTA-treated group, average body weight and the renal organ coefficient were significantly lower (*p* < 0.01) compared with those in the control group (Figure 1A,B). When the mice were treated with ASX, a significant increase in average body weight was observed (*p* < 0.01); however, the renal organ coefficient was not significantly different compared with that of the controls. Furthermore, compared with the OTA group, the mice treated with ASX followed by OTA showed a significant increase in average body weight (*p* < 0.01), but no significant difference in the renal organ coefficient.

Compared with the control group, the levels of UA and BUN in the serum of mice treated with OTA were significantly increased (*p* < 0.01 and *p* < 0.05); however, the level of CRE was not significantly affected. By contrast, the levels of UA, BUN and CRE in serum were not affected by ASX compared with those in the control. In addition, the UA and the BUN levels in the ASX group were significantly lower than those in the OTA group (*p* < 0.01 and *p* < 0.05), whereas the CRE level was not significantly changed. In the ASX and OTA group, the serum UA and BUN levels were significantly decreased (*p* < 0.01 and *p* < 0.05, respectively) compared with those in the OTA group. There was no significant change in CRE levels in the ASX and OTA group (Figure 1C–E.).

### 2.2. Histopathological Changes

In the control cells, the glomerular structure was intact, the renal small cystic lumen was uniform in size, the proximal tubule epithelial cells were closely arranged, the diameter of the lumen was large, the lumen was irregular, and the cell body was large. The cytoplasm was stained deeply with H&E (hematoxylin-eosin staining). The nucleus was large and round, and located at the base of the cell. The distal tubule lumen was large and obvious, the cells were lightly colored, and the nucleus was round and located on the proximal cavity surface (Figure 2A). In contrast to the control cells, after OTA treatment (Figure 2B), the renal tubules were swollen, and the tubular capillaries became narrow and even disappeared. Renal tubular epithelial cells showed granule degeneration, vesicular degeneration, nuclear condensation, and nuclear lysis. No gross morphological changes were seen after ASX treatment (Figure 2C). In the OTA and ASX group (Figure 2D), glomerular swelling was relieved, the renal cystic space was restored, and tubular epithelial cell degeneration and necrosis were reduced.

In the control and ASX groups, intact kidney tissue structures were observed. In contrast, in the OTA-treated group, glomerular swelling and congestion (a), vesicular degeneration of renal tubular epithelial cell (b), pyknosis (c), karyorhexis (d) and karyolysis (e) were observed. The arrows “
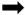
” indicate a pathological injury in the kidney.

### 2.3. Electron Microscopic Observation

Under transmission electron microscopy, the kidney cells were closely arranged and intact in the control group (Figure 3A) and the ASX group (Figure 3C). The mitochondria had a clear structure, the mitochondrial cristae were neatly arranged, and uniform chromatin distribution was observed in the nucleus. In contrast, in the OTA treatment group (Figure 3B), the mitochondrial structure was disordered, the mitochondrial cristae disappeared, the nuclear chromatin was dispersed, the basement membrane was thickened, the nuclear membrane had disappeared, and the endoplasmic reticulum was swollen. In the ASX and OTA group (Figure 3D), the normal appearance of the mitochondria was restored, and the structure of the mitochondrial cristae and the nucleus was nucleus was intact.

In the control and ASX groups, the renal cell structure was observed (b) (g) and the nuclear chromatin was uniform (a) (h). In contrast, in the image of the OTA group, mitochondrial structure disorder, mitochondrial cristae disappearance (c), nuclear chromatin diffusion (f), nuclear membrane disappearance (d), and endoplasmic reticulum swelling were observed (e). In the image of the ASX and OTA group, the mitochondria appeared normal (i), and the structure of mitochondria cristae (i) and the nuclear was intact (j).

### 2.4. Apoptotic Analysis

The results of terminal deoxynucleotidyl transferase nick-end-labeling (TUNEL) staining under fluorescence microscopy showed that compared with that in the control group, the number of apoptotic cells was significantly increased after OTA treatment (*p* < 0.01). No significant difference was observed between the ASX group and ASX and OTA group. The numbers of apoptotic cells in the ASX group and the ASX and OTA group were significantly lower compared with those in the OTA group (*p* < 0.01) (Figure 4A,B).

### 2.5. Antioxidant Status

Compared with the control group, significantly lower levels of total superoxide dismutase (T-SOD), glutathione (GSH), and catalase (CAT) were detected in the kidney tissue of the OTA group (*p* < 0.01) (Figure 5B–D), whereas the malondialdehyde (MDA) levels increased significantly (*p* < 0.01) (Figure 5A). In contrast, CAT levels increased significantly in the ASX group (*p* < 0.01) (Figure 5D). There were no statistical differences in MDA, GSH, and T-SOD levels compared with those in the control group. In addition, compared with those in the OTA group, the GSH, T-SOD, and CAT levels in the ASX and the ASX and OTA groups were increased significantly (*p* < 0.01), whereas a significant decrease in MDA was noted (*p* < 0.01).

### 2.6. Gene Expression Related to the NRF2/KEAP1 Signaling Pathway

After OTA treatment, the mRNA levels of *Nrf2* (encoding nuclear factor, erythroid 2 like), *Nqo*1 (encoding NAD(P)H quinone dehydrogenase), *Ho1* (encoding heme oxygenase 1), *Gclc* (γ-GCS, gamma-glutamylcysteine synthetase), and *Gpx1* (GSH-Px, glutathione peroxidase 1) were significantly decreased compared with those in the control group (*p* < 0.05 or *p* < 0.01). The *Keap1* (encoding Kelch-like Epichlorohydrin-associated protein) mRNA level increased significantly (*p* < 0.01). By contrast, ASX treatment significantly upregulated the mRNA levels of *Nrf2*, *Nqo*1, *Ho1*, *Gclc*, and *Gpx1* compared with those in control group (*p* < 0.05 or *p* < 0.01) and significantly downregulated the mRNA level of *Keap1* (*p* < 0.05). Treatment with ASX and OTA significantly upregulated the mRNA levels of *Nrf2*, *Nqo*1, *Ho1*, *Gclc*, and *Gpx1* compared with those in the OTA group (*p* < 0.05 or *p* < 0.01), and significantly downregulated the mRNA level of *Keap1* (*p* < 0.01) (Figure 6).

Quantitative real-time reverse transcription PCR (qRT-PCR) analysis of the effect of ASX on gene expression of the NRF2/KEAP1 signaling pathway in the renal tissue of the control and experimental mice. Data are shown as the mean ± standard deviation (SD). n = 6 mice/group. ^ *p* < 0.05, ^^ *p* < 0.01 vs. the OTA group; * *p* < 0.05, ** *p* < 0.01 vs. the control group.

### 2.7. Protein Levels of Members of the NRF2/KEAP1 Signaling Pathway

After OTA treatment, the levels of NRF2-related target proteins NQO1, HO-1, γ-GCS, and GSH-Px decreased compared with those in the control group (*p* < 0.05 or *p* < 0.01); however, the levels of KEAP1 increased (*p* < 0.01). After ASX treatment, the levels of NRF2-related target proteins increased in the ASX group and the ASX and OTA group compared with those in the OTA group (*p* < 0.05 or *p* < 0.01), while the level of KEAP1 decreased (*p* < 0.05); the protein levels of NRF2 in the nuclear was significantly increased, and the level of NRF2 in the cytoplasm was significantly downregulated (*p* < 0.05 or *p* < 0.01) (Figure 7).

## 3. Discussion

The obvious toxicity of OTA, and its seemingly common contamination of feed and food, have attracted increased research attention in the fields of food safety and animal feed contamination. There is also an urgent need to identify alternative pathways to limit OTA toxicity. In recent years, the biological effects of ASX have been studied because of its excellent antioxidant properties. In the present study, we explored the protective mechanism of ASX against renal injury caused by OTA.

OTA mainly damages the kidney and liver of animals, and the kidney is the first target organ. Mor et al. found that in 16-week-old rats, after 6 weeks of OTA at 5 mg/kg, their body weight (BW) was slightly decreased compared with that of the control group [36]. Hope et al. found that the urine output of the OTA-exposed group increased compared with that of the healthy control group [37]. After establishing the animal model, we observed symptoms of typical subacute OTA poisoning after OTA treatment, such as depression, loss of appetite, weight loss, dehydration, polydipsia, and polyuria (Appendix A).

Biomarkers in the blood are often used to detect kidney damage. UA, BUN, and CRE levels in serum are indicators of kidney health, and elevated levels of UA, BUN, and CRE indicate impaired renal function [38,39]. Our results showed that after OTA exposure, the mice lost weight, their organ coefficient decreased, and their serum UA and BUN levels increased significantly. These results indicated that the mice in the OTA group had been poisoned. This deduction was confirmed using H&E staining, electron microscopy, and TUNEL staining, which were consistent with previous reports [40,41]. In addition, we found non-significant variations in CRE levels across all the groups due to the fact that OTA mainly affects the renal tubular reabsorption function, but has little effect on glomerular filtration function, and further research is needed to confirm this [42]. There is increasing evidence that OTA causes its toxic effects via oxidative stress [43]. In contrast, after treatment with both ASX and OTA, the levels of the above-mentioned biomarkers were restored towards control levels. Our results showed that ASX could effectively alleviate the renal damage caused by OTA exposure.

MDA is considered the most representative the lipid peroxidation end product and this is often used as a marker for lipid peroxidation [44,45]. In the present study, the MDA content of the OTA-treated group increased significantly, suggesting that, in response to OTA, the body overproduces ROS and undergoes lipid peroxidation reactions, resulting in oxidative stress. By contrast, ASX pretreatment reduced OTA-induced MDA levels in mouse kidneys. This suggested that the protective effect of ASX on mouse kidney injury might be related to the reduction in lipid peroxidation. In vivo antioxidant systems include non-enzymatic systems such as GSH and antioxidant enzyme systems (SOD, GSH-Px, and CAT). The results of the present study showed that GSH, T-SOD, and CAT activities were reduced after OTA exposure. In contrast, after ASX pretreatment, the mice had significantly improved antioxidant enzyme activities (SOD, GSH and CAT), and improved OTA-induced kidney damage. Therefore, the antioxidant effect of ASX seems to play an important role in protecting the kidneys upon exposure to OTA. ASX has a unique molecular structure that exerts its powerful antioxidant activity by quenching singlet oxygen and scavenging free radicals [46,47,48]. Our results are consistent with the antioxidant effects of ASX in organ damage caused by various toxins.

NRF2 is an important transcriptional activator of antioxidant genes and is important in antioxidant protection [49]. After activation of the NRF2 signaling pathway, it contributes to the expression of a series of downstream antioxidant proteins, such as NQO1, HO-1, γ-GCS, and GSH-Px [50]. Previous studies have also shown that grape seed proanthocyanidins reduce the cytotoxicity of cisplatin-induced human embryonic kidney cells by regulating NRF2/HO-1 in vitro [51]. The hibiscus anthocyanins extracted from hibiscus flowers play anti-oxidative stress and cytoprotective roles by activating the NRF2/HO-1 axis to maintain a normal mitochondrial membrane potential and ROS production in the cell [52]. To further demonstrate that ASX may affect the kidney through the NRF2 signaling pathway in OTA-induced oxidative stress, we examined the mRNA and protein expressions of NRF2, NQO1, HO-1, γ-GCS, and GSH-Px. The results showed that, after OTA exposure, the protein levels of NRF2 in the nucleus of kidney cells was increased compared with the control group. This suggested that, regarding the activation of the nuclear transfer of NRF2, ASX pretreatment further promoted the nuclear transfer. ASX activated the NRF2/KEAP1 pathway to counter the toxic effect of OTA on mouse kidneys. Under physiological conditions, the NRF2 expression level is low; however, it can be induced significantly under the influence of factors causing oxidative stress, such as poisoning, volatility, hyperglycemia, and hypoxia [53,54]. Our observations are consistent with the classical pattern of NRF2 activation [50]. In addition, the level of NRF2 in the cytoplasm was significantly downregulated after OTA treatment, and the level of NRF2 in the cytoplasm was even lower after ASX pretreatment. This finding is consistent with previous research [55]. Therefore, our results confirm that ASX promotes the transfer of NRF2 from the cytoplasm to the nucleus, resulting in enhanced expression of its downstream targets NQO1, HO1, γ-GCS, and GSH-Px, which improves the redox balance, increases the protection of mouse kidney cells, and ultimately improves the body’s antioxidant capacity (Figure 8). Furthermore, under long-term exposure to OTA, oxidative stress reactions occurred in the kidneys of mice and intracellular NQO1, HO-1, γ-GCS, and GSH-Px were consumed in large quantities to reduce OTA-induced oxidative stress. The results showed that the expressions of NQO1, HO-1, γ-GCS, and GSH-Px in the kidneys of the OTA-treated mice were significantly downregulated. The results of the present study indicate that ASX could improve OTA-induced oxidative damage by activating the NRF2 signaling pathway to protect the kidneys of C57BL/J mice.

## 4. Materials and Methods

### 4.1. Chemicals and Drugs

OTA was purchased from LKT Labs. Inc. (St Paul, MN, USA). ASX of ester form, extracted from *Haematococcus pluvialis* (purity > 98%), was bought from Beijing Solarbio Science & Technology Co. Ltd., Beijing, China. For oral administration, OTA was dissolved in 0.1 mol/L NaHCO_3_ [56] and ASX was dissolved in olive oil at 1 kg/L [57]. All other reagents used in the study were of analytical grade.

### 4.2. Animals

Six-week-old C57BL/J mice weighing 20 ± 2 g were purchased from Shandong Peng Yue Experimental Animal Breeding Co. Ltd. (Shandong, China). The mice were reared in a specific pathogen-free environment at 22 ± 2 °C, with a relative humidity of 50 ± 1%, and a 12/12 h light/dark cycle. Feed and water were provided *ad libitum*. The mice were allowed to acclimate for three weeks before the experiment. The study was performed with the approval of the Ethics Committee for Laboratory Animal Care (Animal Ethics Procedures and Guidelines of China) for use by Shenyang Agricultural University, China (Permit No. 264 SYXK<Liao>2011-0001, 20, October 2011).

### 4.3. Experimental Design

The C57BL/J mice were weighed and randomly divided into four groups (n = 20 per group): group I (Control), group II (OTA), group III (ASX), and group IV (ASX and OTA). All treatments were administered orally. Group I received 0.1 mL of olive oil and then gavage with 0.1 mL NaHCO_3_ after 2 h. Group II received OTA at 5 mg/kg body weight (BW)/day and then 0.1 mL olive oil after 2 h [58]. Group III received ASX at 100 mg/kg BW/day and then 0.1 mL NaHCO_3_ after 2 h [59]. Group IV received 100 mg/kg BW/day of ASX and 2 h later they received 5 mg/kg BW/day of OTA. The treatments were administered for 7 days a week and then discontinued for 2 days, repeated three times. Dietary intake and BW were measured every day over the course of the study period.

### 4.4. Sample Preparation

After 27 days of treatment, the animals were weighed, and then sacrificed humanely of cervical dislocation (according to the 2013 AVMA Guideline for the Euthanasia of Animals) [60]. Blood samples were obtained and centrifuged in a cryogenic centrifuge (Thermo Fisher Scientific, Waltham, MA, USA) at 3000× *g* at 4 °C for 10 min. The supernatant (serum) was collected for analysis and subsequent testing for biochemical indicators. The mice were sacrificed quickly after blood collection, and their kidneys were excised and weighed. The renal organ coefficient was calculated as the organ weight divided by the body weight. The left-side kidneys were fixed in 4% paraformaldehyde and stored for further histopathological analysis. One hundred milligrams of right-side kidneys tissue was placed into a pre-cooled homogenizer with the correct volume of saline [tissue weight (g):saline volume (mL) = 1:9], and ground to provide a tissue homogenate. The homogenate was centrifuged at 3000× *g* for 10 min at 4 °C and the supernatant was retained for subsequent analysis. The remaining portion of the right-side kidneys were stored at −80 °C for further use.

### 4.5. Measurement of Serum Biochemical Indices

Serum levels of uric acid (UA), blood urea nitrogen (BUN), and creatinine (CRE) were determined using a standard commercial kit (Changchun Huili Biotechnology Co. Ltd., Changchun, China) according to the manufacturer’s instructions, and were detected using a fully automatic biochemical analyzer (Chemray 240; Rayto Life and Analytical Sciences Co. Ltd., Shenzhen, China).

### 4.6. Histopathology of the Kidney

After 48 h of fixation in 4% paraformaldehyde, the kidney tissue was washed with saline, dehydrated in an ascending series of ethanol concentrations, dipped in paraffin wax, paraffin-embedded, sectioned (4 μm thickness) (RM2016, Leica, Beijing, China), stained with hematoxylin and eosin (H&E) (Servicebio, Wuhan, China), and then observed under an optical microscope (Leica DM750, Leica, Beijing, China).

### 4.7. Electron Microscopy Observation

Kidney tissue was cut into 1 mm sized tissue sections and then placed in a 4 °C pre-cooled 0.25% glutaraldehyde fixative for 24 h. After rinsing (three times with 1 M PBS buffer pH = 7.2 for 15 min each time), post-fixing with 1–2% osmium acid for 2 h, rinsing (three times with 1 M PBS buffer pH = 7.2, 15 min each time), dehydration (using 30%, 50%, and 70% ethanol, 15 min each time, and then three times with 100% ethanol for 30 min each time), dehydration (using 80%, 90%, 95% acetone, 15 min each time, and then three times with 100% acetone for 10 min each time) embedding (in epoxy resin (EP8012)), polymerization and hardening (40 °C and 60 °C, 48 h each), sectioning (thickness 70–90 nm), and staining (2% uranium acetate and 6% lead citrate, 10 min each), the sections were imaged under a transmission electron microscope (Hitachi HT7700. Hitachi, Tokyo, Japan).

### 4.8. Observation of Apoptotic Cells

According to the manufacturer’s instructions, a TUNEL analytical kit was used (Roche, Basel, Switzerland). The paraffin sections for TUNEL analysis were produced by China Seville analytical Biotechnology Co. Ltd., Wuhan, China. After TUNEL staining, DAPI (2-(4-amidinophenyl)-1H-indole-6-carboxamidine) was used to stain the nuclei.

### 4.9. Antioxidant Status

A BCA commercial kit (Nanjing Jiancheng Bioengineering Institute, Nanjing, China) was used to assess the protein concentration of fresh 10% tissue homogenate supernatant samples. Commercial kits (Nanjing Jiancheng Bioengineering Institute) were used to determine the antioxidant status of the C57BL/J mice based on their glutathione (GSH) and malondialdehyde (MDA) contents, and the activities of total superoxide dismutase (T-SOD) and catalase (CAT). The assays were performed according to the manufacturer’s guidelines.

### 4.10. Analysis of Gene Expression

A Total RNA Extraction Kit (Sangon Biotech Co. Ltd. Shanghai, China) was used to extract total RNAs from kidney tissue, according to the manufacturer’s instructions. Total RNA purity was assessed based on the OD 260/280 nm ratio. The total RNA was used to synthesize cDNA according to the manufacturer’s guidelines of the Prime Script RT Master Mix kit (Takara Co. Ltd. Dalian, China). Quantitative real-time reverse transcription PCR (iQ5, ABI, Waltham, MA, USA) was used to detect the mRNA expression levels of *Nrf2*, *Keap1*, *Nqo1*, *Ho1*, *Gclc*, and *Gpx1* in the kidneys, according to the method of He et al. [61]. The qRT-PCR data was processed using the 2^−ΔΔCt^ method [62]. All expression values were normalized to that of the *Actb* (β-actin) gene. Table 1 shows the PCR primers, which were synthesized by Sangon Biotech Institute Co. Ltd.

### 4.11. Western Blotting Analysis

Kidney tissue (30 mg) was added to 200 µl of Radio immuneprecipitation assay (RIPA) buffer containing 1 mM of phenylmethylsulfonyl fluoride and the protease inhibitor PMSF (Sangon Biotech Co. Ltd. Beijing, China), and centrifuged at 4 °C and 12,000× *g*. The supernatant was collected and its protein concentration was quantified using the BCA kit (Sangon Biotech Co. Ltd. Beijing, China). Nuclear protein and cytoplasmic proteins were extracted according to the manufacturer’s instructions for the respective kits (Active Motif Inc., Carlsbad, CA, USA). Protein samples were subjected to 10% SDS-PAGE and then transferred to nitrocellulose membranes using a Trans-Blot machine (Bio-Rad, Hercules, CA, USA). The membranes were blocked using tris-buffered saline with tween-20 (TBST) containing 5% bovine serum albumin (BSA) for 2 h at 37 °C. The membranes were then incubated with primary antibodies against NRF2 (1:2000), KEAP1 (1:2000), NQO1 (1:4000), HO-1 (1:4000), γ-GCS (1:4000), GSH-Px (1: 4000), and Beta actin (1:5000) (all from Abcam, Tokyo, Japan); and nuclear reference Histone 3 (Cell Signaling Technology, Boston, MA, USA) diluted in phosphate-buffered saline Tween-20 (PBST) at 4 °C overnight. The membranes were washed three times with PBST, incubated with horseradish peroxidase (HRP)-conjugated secondary antibody (1:5000) (Abcam), and then developed using an ECL chemiluminescence solution (Solarbio Science & Technology Co. Ltd., Beijing, China) before being exposed to X-ray films. The DNR bioimaging system (Neve Yamin, Israel) was used to determine the relative intensity of the immunoreactive protein bands.

### 4.12. Statistical Analysis

Data are expressed as the mean ± standard deviation (SD). SPSS software version 18.0 (IBM Corp., Armonk, NY, USA) was used to carry out all the statistical tests. One-way analysis of variance (ANOVA) was used to evaluate significant differences among the multiple groups as a post-hoc test. Statistical significance was considered at *p* < 0.05.

## 5. Conclusions

The results of the present study showed that ASX protects against OTA-induced renal injury. ASX’s mechanism of action involves regulating the NFR2/KEAP1 signaling pathway. This study provides a new theoretical basis and directions for the clinical application of ASX to prevent and treat oxidative damage to organs and feed additives.

## Figures and Tables

**Figure 1 molecules-25-01386-f001:**
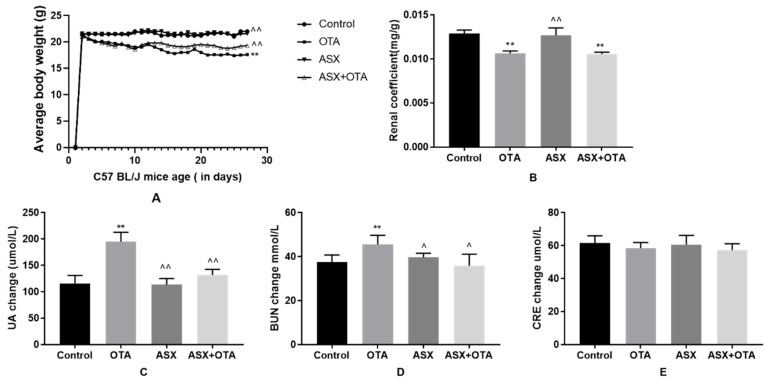
Assessment of mouse average body weight, organ coefficients and changes in serum biochemical indicators. (**A**) Average body weight change of C57BL/J from day 1 to day 27, (**B**) renal organ coefficients, (**C**) changes in uric acid (UA) levels, (**D**) changes in blood urea nitrogen (BUN) levels, and (**E**) changes in creatinine (CRE) levels in the different groups. Values represent the mean ± standard deviation (SD). n = 8 mice/group. ^ *p* < 0.05, ^^ *p* < 0.01 vs. the ochratoxin A (OTA) group; * *p* < 0.05, ** *p* < 0.01 vs. the control group.

**Figure 2 molecules-25-01386-f002:**
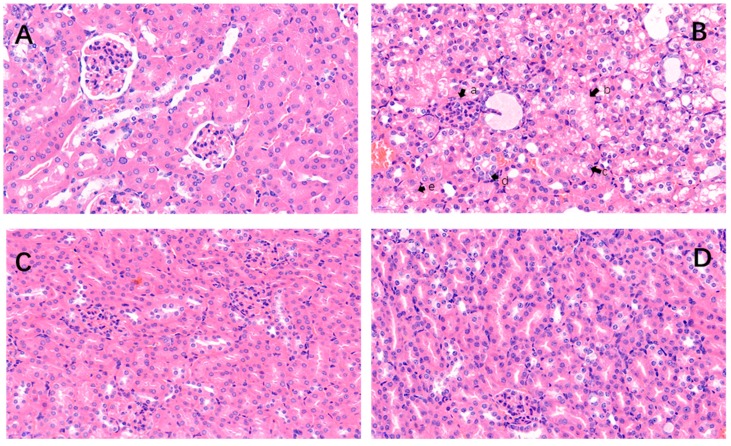
Histopathological changes. Renal micrographs of hematoxylin and eosin (H&E) staining (400×) of the control (**A**), OTA (**B**), astaxanthin (ASX) (**C**), ASX and OTA groups (**D**). n = 6 mice/group.

**Figure 3 molecules-25-01386-f003:**
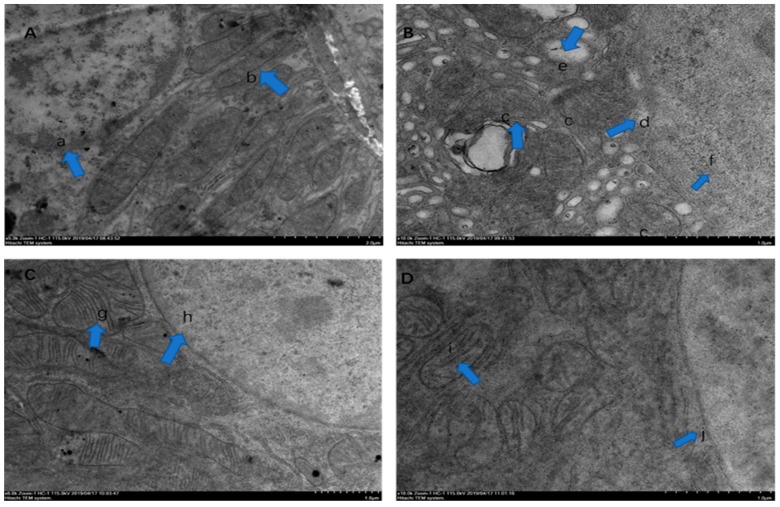
Electron microscopy observation. Renal electron micrographs of the control (**A**), OTA (**B**), ASX (**C**), ASX and OTA (**D**) groups. n = 6 mice/group.

**Figure 4 molecules-25-01386-f004:**
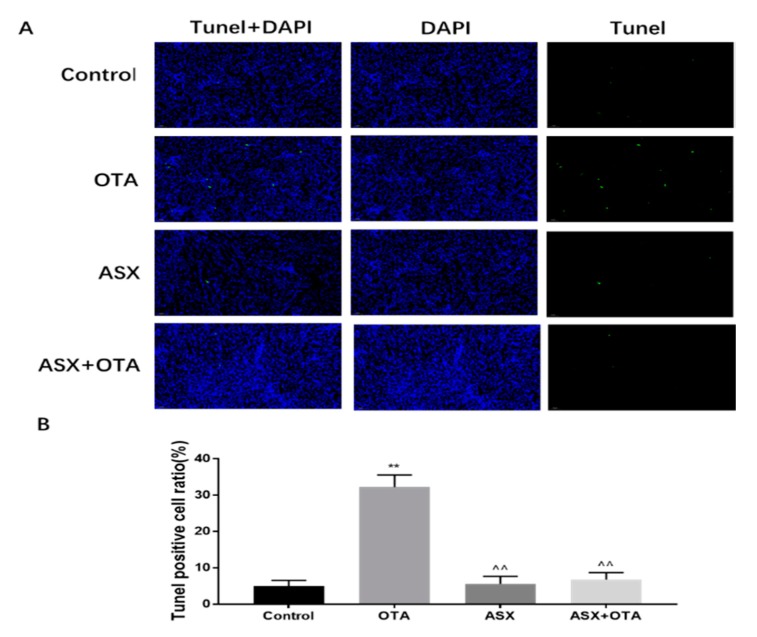
Apoptosis analysis. (**A**) Renal terminal deoxynucleotidyl transferase nick-end-labeling (TUNEL) staining (200×). Control, OTA, ASX, ASX and OTA groups. Green fluorescence represents apoptotic cells stained by TUNEL. DAPI was used for nuclear staining. (**B**) TUNEL positive cell rate. ^ *p* < 0.05, ^^ *p* < 0.01 vs. the OTA group; * *p* < 0.05, ** *p* < 0.01 vs. the control group. n = 6 mice/group.

**Figure 5 molecules-25-01386-f005:**
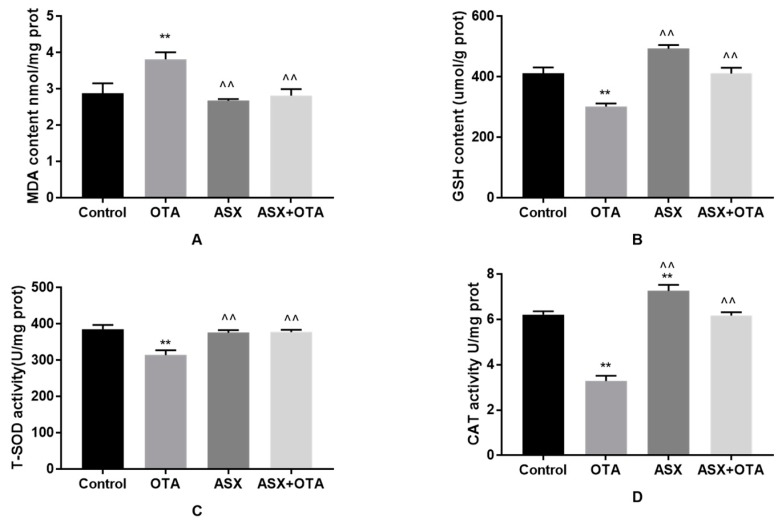
Antioxidant status. The contents of malondialdehyde (MDA) (**A**) and glutathione (GSH) (**B**), and the total superoxide dismutase (T-SOD) (**C**) and catalase (CAT) (**D**) levers in the renal tissue in the four groups. Values represent the mean ± the standard deviation (SD). group. n = 7 mice/group. ^ *p* < 0.05, ^^ *p* < 0.01 vs. the OTA group; * *p* < 0.05, ** *p* < 0.01 vs. the control group.

**Figure 6 molecules-25-01386-f006:**
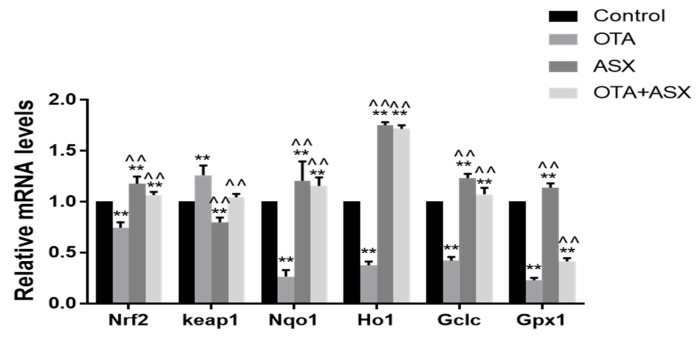
Gene expression in the NRF2/KEAP1 signaling pathway.

**Figure 7 molecules-25-01386-f007:**
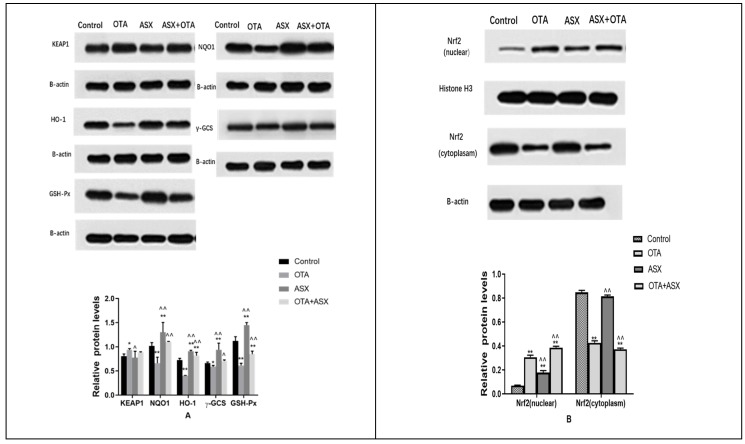
Protein levels of members of the NRF2/KEAP1 signal pathway. (**A**) Western blotting analysis of the effect of ASX on NQO1, HO-1, γ-GCS, GSH-Px, and KEAP1 levels in renal tissues of the control and experimental mice. (**B**) Western blotting analysis of the effect of ASX on nuclear NRF2 and cytoplasmic NRF2 in renal tissues of the control and experimental mice. Protein levels were normalized against those of β-actin and histone. Data are shown as the mean ± standard deviation (SD). n = 6 mice/group. ^ *p* < 0.05, ^^ *p* < 0.01 vs. the OTA group; * *p* < 0.05, ** *p* < 0.01 vs. the control group.

**Figure 8 molecules-25-01386-f008:**
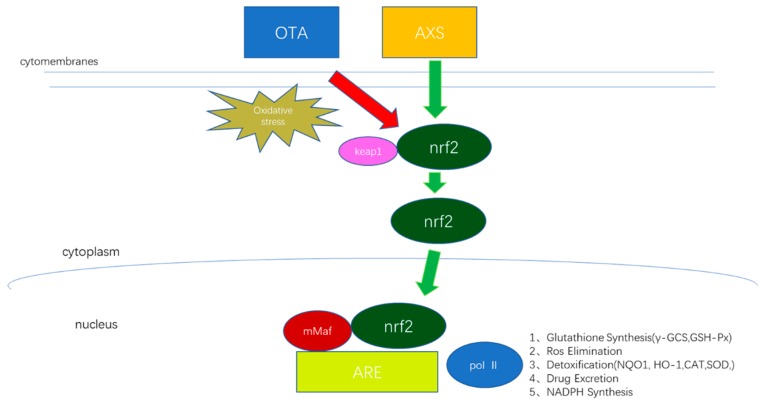
The flow diagram of the NRF2/KEAP1 signal pathway. Under physiological conditions, Nrf2 is negatively regulated and ubiquitinated though Keap1. When oxidative stress occurs, the interaction between Nrf2 and Keap1 is destroyed, Nrf2 of free type accumulates in the cytoplasm, Nrf2 translocations to the nucleus were increased, and this activates downstream antioxidant protein to resist oxidative stress.

**Table 1 molecules-25-01386-t001:** The sequences of primers used to amplify the target genes.

Primer	Sequence (5′-3′)	Melting Temperature (°C)	Size (bp)
M-Nrf2-F	CCTATGCGTGAATCCCAAT	58.7	120
M-Nrf2-R	TGTGAGATGAGCCTCTAAGCG	58.1	
M-Keap1-F	ATGTTGACACGGAGGATTGG	57.7	133
M-Keap1-R	TCATCCGCCACTCATTCCT	58.2	
M-Nqo1-F	GCGAGAAGAGCCCTGATTGT	59.2	177
M-Nqo1-R	CTTCAGCTCACCTGTGATGTCAT	59.2	
M-Ho1-F	CAAGCCGAGAATGCTGAGTT	57.9	105
M-Ho1-R	CAGGGCCGTGTAGATATGGTA	57.8	172
M-Gclc-F	TCGCCTCCGATTGAAGATG	59.1	
M-Gclc-R	TACTATTGGGTTTTACCTGTGCC	58.3	
M-Gpx1-F	AGGAGAATGGCAAGAATGAAGA	58.3	136
M-Gpx1-R	AGGAAGGTAAAGAGCGGGTG	58.6	174
M-b-actin-F	GTGCTATGTTGCTCTAGACTTCG	56.8	
M-b-actin-R	ATGCCACAGGATTCCATACC	56.9	

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
