# Peer review of "Protective Effect of Astaxanthin on Ochratoxin A-Induced Kidney Injury to Mice by Regulating Oxidative Stress-Related NRF2/KEAP1 Pathway"

_molecules, 2020, doi:10.3390/molecules25061386_

Round 1

Reviewer 1 Report

Manuscript title: Protective effect of Astaxanthin on Ochratoxin A-induced kidney injury to the mice by Regulating Oxidative Stress-Related NRF2/KEAP1 pathway In the manuscript, the authors present the protective effects of astaxanthin (ASX) on ochratoxin A (OTA)-induced kidney injury in mice. In general, the authors have completed a reasonable study with very informative data on the mechanism of action of ASX on the preventive activity from the kidney injury of OTA. However, some of minor concerns would be suggested and requested for further improvement in the manuscript. 1. It was seemed unclearly whether the ASX, extracted from Haematococcus pluvialis (purity > 98%), was free form or the ester form. Because of the ester form of ASX existed mostly in the algae, we should carefully apply the sample before the confirmation. 2. Why the dosage of ASX was decided to apply in the study? The dosage of 100 mg/Kg BW seemed to be quite a large amount. Ten mg would be quite effective dosage in prevention study as previous report (doi.org/10.3390/ijms17071128). 3. The finding of the paper depends on the expression targeted proteins. It is not clear whether authors have presented clearly with the pathway of active mechanisms in the study. Therefore, the brief flow diagram of the pathway would be suggested shown in the Results. 4. Please revised the format of references in according to the regulation of the Journal.

Reviewer 2 Report

The manuscript submitted by Dr Li L et al deals with the putative therapeutic effect of Astaxanthin (ASX) on Ochratoxin A (OTA) induced kidney injury.

Because the mycotoxin OTA is known to be one of the most abundant toxin contaminating food products and consequently cause alteration of human health this study is of interest. The authors have investigated the putative protective effect of ASX (known as a powerful antioxidant) on renal function and tissue by using a mouse model treated with OTA by oral route for 27 days.

The data obtained show that ASX could improve OTA-induced oxidative damage by activating the NRF2 signaling pathway to protect mice kidneys.

The design of the study is straightforward and the data are clearly presented but deserve additional experiments to be more convincing.

Major comments

-How were OTA and ASX doses and treatment period determined?

-Specify the LD50 of OTA.

-Because of gavage administration and thus microbiote interaction one can wonder what is the blood concentration of OTA and ASX.

-Do the authors have any evidence of OTA and ASX presence at the kidney level?

-To better appreciate the nephrotoxicity of OTA and consequently the protective effect of ASX a better characterization/quantification of renal function and histological damage should be performed.

The H&E images are not really convincing. Kidney sections stained with PAS should be shown (PAS positive material should be semi-quantified) as well as the semi-quantification of glomerular swelling (volume) and tubular alteration (atrophy).

Do the authors observe any mononuclear cells infiltration?

-The authors claimed in the discussion section that OTA induced loss of appetite, polydipsia and polyuria. The data have to be shown (at least in suppl data).

-Because you observed weight loss and dehydration BUN and CRE in serum are not very good markers to assess renal function.  Renal function (glomerular filtration rate) should be assessed by a better recognized evaluation method i.e inulin/Cistatin C clearance.  Other biomarkers (e.g Kim 1, NGAL, IL18) of kidney damage should be assessed.

-Since the authors have performed electron microscopy, do they observe any variation in foot process effacement following OTA administration?

-Please precise how the TUNEL quantification was performed.

Minor comments

Title: replace Rejulating by Regulating

Results section lines 189-196: please specify (Fig 7) in the text.

Discussion line 257 replace “pathway to against” by “pathway to counter”

To better appreciate statistical dispersion scatter dot plot should replace bar-histogram

Reviewer 3 Report

The study conducted by Li et. demonstrated the probable protective role of astaxanthin on ochratoxin A-mediated renal damage in a C57BL/J mice model. The rationale for the work was clearly justified with the experimentals adequately planned and executed with specific emphasis on renal biomarkers and intrinsic pathway of interest. The results were also well presented and justifiable reasons given in the discussion section for the results obtained. However, the following needs to be amended prior to further consideration;

  1.  Provide suitable reference for the dose of ASX used in section 4.3 and clearly state the method of humane sacrificing used for the animals in section 4.4. Please, provide suitable reference as well
  2.  Harmonize or clarify the number of animals (n) used, particularly in some of the figures presented. For instance, some had n=8 as in Fig 1, n=6 as in Figs 6 and 7,  
  3. Correct the spelling 'rejulating' to 'regulating' in the topic and 'nulceotidyl' to 'nucleotidyl' in section 2.4. Carefully read through the whole manuscript to correct other similar errors
  4.  Use 'activity' for SOD and CAT but level for MDA and GSH throughout the entire manuscript
  5.  Remove the last three statements of the 'Introduction' section (Lines 75-80) and merge with an appropriate section of the discussion or conclusion
  6.  Please, complete and be explicit with the statement in Lines 163-164
  7.  Delete the 'of' in Line 170
  8.  Justify the reason for the non-significant variations in CRE levels across all the groups as observed in this study (Lines 95-98 and Fig 1E). The reason for this observation must be included in the discussion section
